# TransformerFusion: Monocular RGB Scene Reconstruction using Transformers

Aljaž Božič [1]    Pablo Palafox [1]    Justus Thies [1,2]    Angela Dai [1]    Matthias Nießner [1]

[1]Technical University of Munich
[2]Max Planck Institute for Intelligent Systems, Tübingen, Germany
aljazbozic.github.io/transformerfusion

## Abstract

We introduce TransformerFusion, a transformer-based 3D scene reconstruction approach. From an input monocular RGB video, the video frames are processed by a transformer network that fuses the observations into a volumetric feature grid representing the scene; this feature grid is then decoded into an implicit 3D scene representation. Key to our approach is the transformer architecture that enables the network to learn to attend to the most relevant image frames for each 3D location in the scene, supervised only by the scene reconstruction task. Features are fused in a coarse-to-fine fashion, storing fine-level features only where needed, requiring lower memory storage and enabling fusion at interactive rates. The feature grid is then decoded to a higher-resolution scene reconstruction, using an MLP-based surface occupancy prediction from interpolated coarse-to-fine 3D features. Our approach results in an accurate surface reconstruction, outperforming state-of-the-art multi-view stereo depth estimation methods, fully-convolutional 3D reconstruction approaches, and approaches using LSTM- or GRU-based recurrent networks for video sequence fusion.

## 1   Introduction

Monocular 3D reconstruction is a core task in 3D computer vision, aiming to reconstruct a complete and accurate 3D geometry of an object or an environment from only 2D observations captured by an RGB camera. A geometric understanding is key to applications such as robotic or autonomous vehicle navigation or interaction, as well as model creation and scene editing for augmented and virtual reality. In addition, geometric scene reconstructions form the basis for 3D scene understanding, supporting tasks such as 3D object detection, semantic, and instance segmentation [34, 35, 36, 29, 7, 43, 15, 16].

While state-of-the-art SLAM systems [3, 41] achieve robust and scale-accurate camera tracking leveraging both visual and inertial measurements, dense and complete 3D reconstruction of large-scale environments from monocular video remains a very challenging problem – particularly for interactive settings. Simultaneously, notable progress has been made on multi-view depth estimation, estimating depth from pairs of images by averaging features extracted from the images in a feature cost volume [42, 17, 19, 38, 13]. Unfortunately, averaging features across a full video sequence can lead to equal-weight treatment of each individual frame, despite some frames possibly containing less information in various regions (e.g., from motion blur, rolling shutter artifacts, very glancing or partial views of objects), making high-fidelity scene reconstruction challenging.

Inspired by the recent advances in natural language processing (NLP) that leverage transformer-based models for sequence to sequence modelling [40, 11, 2], we propose a transformer-based method that fuses a sequence of RGB input frames into a 3D representation of a scene at interactive rates. Key to

35th Conference on Neural Information Processing Systems (NeurIPS 2021).

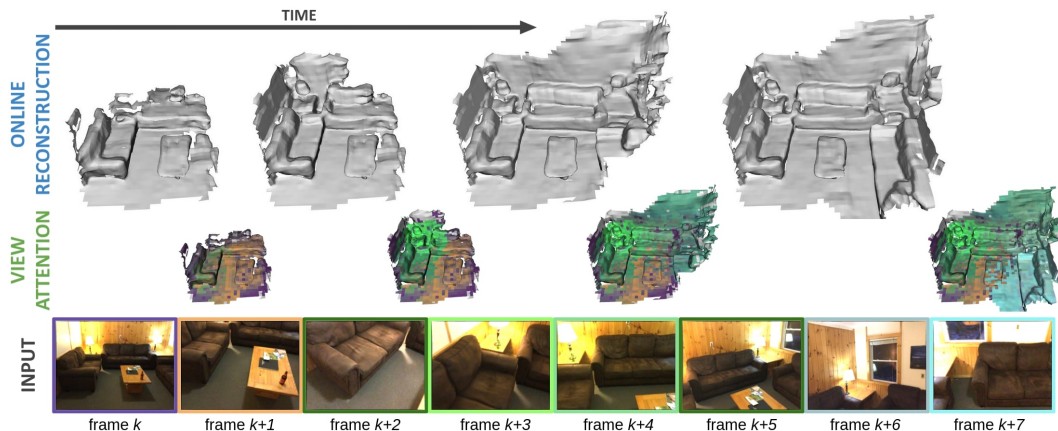

Figure 1: TransformerFusion is an online scene reconstruction method that takes a monocular RGB video as input. The features extracted from each observed image are fused incrementally with a transformer architecture. This fusion approach learns to attend to the most relevant image frames for each 3D location (see view attention color maps of the most relevant frame) achieving state-of-the-art reconstruction results.

our approach is a learned feature fusion of the video frames using a transformer-based architecture, which learns to attend to the most informative image features to reconstruct a local 3D region of the scene. A new observed RGB frame is encoded into a 2D feature map, and unprojected into a 3D volume, where our transformer learns a fused 3D feature for each location in the 3D volume from the image view features. This enables extraction of the most informative view features for each location in the 3D scene. The 3D features are fused in coarse-to-fine fashion, providing both improved reconstruction performance as well as interactive runtime. These features are then decoded into high-resolution scene geometry with an MLP-based surface occupancy prediction.

In summary, our main contributions to achieve robust and accurate scene reconstructions are:

- Learned multi-view feature fusion in the temporal domain using a transformer network that attends to only the most informative features of the image views for reconstructing each location in a scene.
- A coarse-to-fine hierarchy of our transformer-based feature fusion that enables an online reconstruction approach running at interactive frame-rates.

## 2   Related Work

**Multi-view depth estimation.**   Estimating depth from multi-view image observations has been long-studied in computer vision. COLMAP [37] introduced a patch matching based approach which achieves impressive accuracy and remains established as one of the most popular methods for multi-view stereo. While COLMAP offers robust depth estimation for distinctive features in images, the patch matching struggles to densely reconstruct areas without many distinctive color features, such as floor and walls. Recently, learning-based approaches that build data-driven priors from large-scale datasets have improved depth estimation in these challenging scenarios. Some proposed methods rely only on a 2D network with multiple images concatenated as input [42]. Several recent approaches instead build a shared 3D feature cost volume in reference camera space using feature averaging [13, 17, 19, 25, 26]. These approaches estimate the reference frame's depth within a local window of frames, but some also propagate information from previously estimated depth maps by using probabilistic filtering [25], a Gaussian process [17], or an LSTM bottleneck layer [13]. Such multi-view depth estimation approaches predict single-view depth maps, which must be fused together to construct a geometric 3D representation of the observed scene.

**3D reconstruction from monocular RGB input.**   Multi-view depth estimation approaches can be combined with depth fusion approaches, such as volumetric fusion [6], to obtain a volumetric

reconstruction of the observed scene. MonoFusion [33] is one of the first methods using depth estimate from a real-time variant of PatchMatch stereo [1]. However, fusing noisy depth estimates causes artifacts in the 3D reconstruction, which lead to the development of recent approaches that directly predict the 3D surface reconstruction instead of per-frame depth estimates. One of the first approaches to predict 3D surface occupancy from two input RGB images is SurfaceNet [20], which converts volumetrically averaged colors into 3D surface occupancies using a 3D convolutional network. Atlas [28] extends this approach to a multi-view setting, while also leveraging learned features instead of colors. Recently, NeuralRecon [39] proposed a real-time 3D reconstruction framework, adding GRU units distributed in 3D to fuse reconstructions from different local windows of frames. Our approach also fuses together learned features from RGB frame input in an online fashion, but our transformer-based multi-view feature fusion enables relying only on the most informative features from the observed frames for a particular spatial location in the reconstructed scene, producing more accurate 3D reconstructions.

**Transformers in computer vision.** The transformer architecture [40] has achieved profound impact in many computer vision tasks in addition to its natural language processing origins. For a detailed survey, we refer the reader to [22]. In computer vision, transformers have been leveraged successfully for tasks such as object detection [4], video classification [44], image classification [12], image generation [30], and human reconstruction [45]. In this work, we propose transformer-based feature fusion for 3D scene reconstruction from a monocular video. Given a sequence of observed RGB frames, our approach learns to attend to the most informative features from each image to predict a dense occupancy field.

## 3 End-to-end 3D Reconstruction using Transformers

Given a set of $N$ RGB images $\mathrm{I}_i \in \mathbb{R}^{W \times H \times 3}$ of a scene with corresponding camera intrinsic parameters $\mathbf{K}_i \in \mathbb{R}^{3 \times 3}$ and extrinsic poses $\mathbf{P}_i \in \mathbb{R}^{4 \times 4}$, our method reconstructs the scene geometry by predicting occupancy values $o \in [0, 1]$ for every 3D point in the scene. Fig. 2 shows an overview of our approach. Each input image $\mathrm{I}_i$ is processed by a 2D convolutional encoder $\Theta$, extracting coarse and fine image features ($\Phi_i^c$ and $\Phi_i^f$, respectively):

$$\Theta : \mathrm{I}_i \in \mathbb{R}^{W \times H \times 3} \mapsto (\Phi_i^c, \Phi_i^f)$$

From these 2D image features, we construct a 3D feature grid in world space. To this end, we regularly sample grid points in 3D at a coarse resolution of every $v_c = 30$ cm and a fine resolution of $v_f = 10$ cm. For these coarse and fine sample points, we query corresponding 2D features in all $N$ images and predict fused coarse $\psi^c$ and fine 3D features $\psi^f$ using transformer networks [40]:

$$\mathcal{T}_c : (\Phi_1^c, \dots, \Phi_N^c) \mapsto (\psi^c, w^c)$$
$$\mathcal{T}_f : (\Phi_1^f, \dots, \Phi_N^f) \mapsto (\psi^f, w^f)$$

Note that we also store the intermediate attention weights $w^c$ and $w^f$ of the first transformer layers for efficient view selection, which is explained in Sec. 3.4.

To further improve the features in the 3D spatial domain, we apply 3D convolutional networks $\mathcal{C}_c$ and $\mathcal{C}_f$, at the coarse and fine level, respectively:

$$\mathcal{C}_c : \{\psi^c\}_{C \times C \times C} \mapsto \{\tilde{\psi}^c\}_{C \times C \times C}$$
$$\mathcal{C}_f : \{(\tilde{\psi}^c, \psi^f)\}_{F \times F \times F} \mapsto \{\tilde{\psi}^f\}_{F \times F \times F}$$

Finally, to predict the scene geometry occupancy for a point $\mathbf{p} \in \mathbb{R}^3$, the coarse $\tilde{\psi}_c$ and fine features $\tilde{\psi}_f$ are trilinearly interpolated and a multi-layer perceptron $\mathcal{S}$ maps these features to occupancies:

$$\mathcal{S} : (\tilde{\psi}^c, \tilde{\psi}^f) \mapsto o \in [0, 1]$$

This extraction of surface occupancies is inspired by convolutional occupancy networks [32] and IFNets [5]. From this occupancy field we extract a surface mesh with Marching cubes [27]. Note that in addition to surface occupancy, we also predict occupancy masks for near-surface locations at the coarse and fine levels. These masks are used for coarse-to-fine surface filtering (see Sec. 3.2), which

improves reconstruction performance with a focus on the surface geometry prediction and enables interactive runtime.

We train our approach in end-to-end fashion by supervising the surface occupancy predictions using the following loss:

$$\mathcal{L} = \mathcal{L}_c + \mathcal{L}_f + \mathcal{L}_o,$$

where $\mathcal{L}_c$ and $\mathcal{L}_f$ denote binary cross-entropy (BCE) losses on occupancy mask predictions for near-surface locations at the coarse and fine levels, respectively (see Sec. 3.2), and $\mathcal{L}_o$ denotes a BCE loss for surface occupancy prediction (see Sec. 3.3).

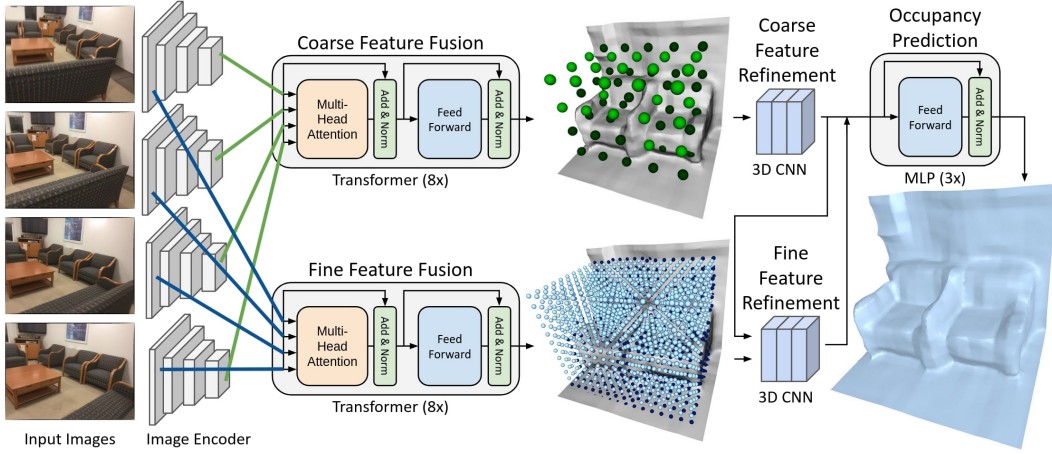

Figure 2: Method overview: given multiple input images, we compute coarse and fine level features. Using a transformer architecture, we separately fuse these coarse and fine features in a voxel grid. To improve the spatial features, we use a refinement network for both the coarse and the fine features. From these feature grids, we extract an occupancy field using a lightweight MLP.

## 3.1 Learning Temporal Feature Fusion via Transformers

For a spatial location $\mathbf{p} \in \mathbb{R}^3$ in the scene reconstruction, we learn to fuse coarse $\psi^c$ and fine level features $\psi^f$ from the $N$ coarse and fine feature images ($\Phi_i^c$ and $\Phi_i^f$, respectively), which are extracted by the 2D encoder $\Theta$. Specifically, we train two instances of a transformer model, one for fusing coarse-level features $\psi^c$ and one for fusing fine-level features $\psi^f$. Both transformers $\mathcal{T}_c$ and $\mathcal{T}_f$ share the same architecture. Thus, for simplicity, we omit the coarse and fine notation in the following.

Our transformer model $\mathcal{T}$ is independently applied to each sample point in world space. For a point $\mathbf{p}$, the transformer network takes a series of 2D features $\phi_i$ as input that are bilinearly sampled from the feature maps $\Phi_i$ at the corresponding projective image location. The projective image location is computed via a full-perspective projection $\Pi_i(\mathbf{p}) = \pi(\mathbf{K}_i(\mathbf{R}_i\mathbf{p} + \mathbf{t}_i))$, assuming known camera intrinsics $\mathbf{K}_i$ and extrinsics $\mathbf{P}_i = (\mathbf{R}_i, \mathbf{t}_i)$. To inform the transformer about invalid features (i.e., a sample point is projected outside an image), we also provide the pixel validity $v_i \in \{0, 1\}$ as input. In addition to these 2D features $\phi_i$, we concatenate the projected depth $d_i = (\mathbf{R}_i\mathbf{p} + \mathbf{t}_i)_z$, and the viewing ray $\mathbf{r_i} = (\mathbf{p} - \mathbf{c}_i)/||\mathbf{p} - \mathbf{c}_i||_2$ to the input ($\mathbf{c}_i \in \mathbb{R}^3$ denoting the camera center of view $i$). These input features are converted to an embedding vector $\theta_i \in \mathbb{R}^D$ using a linear layer $\theta_i = \mathbf{FCN}(\phi_i, d_i, v_i, \mathbf{r_i})$, before feeding it into the transformer network that then predicts a fused feature $\psi \in \mathbb{R}^D$:

$$\mathcal{T} : (\theta_1, \ldots, \theta_N) \mapsto (\psi, w)$$

As described above, $w$ denotes the attention values of the initial attention layer, which are used for view selection to speed-up fusion (see Sec. 3.4).

**Transformer architecture.** We followed [12] when designing the transformer architecture $\mathcal{T}$. It consists of 8 modules of feed-forward and attention layers, using multi-head attention with 4 attention heads and embedding dimension $D = 256$. Feed-forward layers process the temporal inputs independently, and contain ReLU activation, linear layers with residual connection, and layer norm.

The model returns both fused feature $\psi \in \mathbb{R}^D$ and attention weights $w \in \mathbb{R}^N$ over all temporal inputs from the initial attention layer that are later used for selecting which views to maintain over longer sequences of input image views.

## 3.2 Spatial Feature Refinement

While the transformer network fuses 2D observations in the temporal domain, we additionally imbue explicit spatial reasoning by applying a 3D CNN to spatially refine the fused features $\{\psi^c\}_{C \times C \times C}$ and $\{\psi^f\}_{F \times F \times F}$ that are computed by the transformers $\mathcal{T}_c$ and $\mathcal{T}_f$ on the coarse and fine grid, respectively. The coarse features $\{\psi^c\}_{C \times C \times C}$ are refined by a 3D CNN $\mathcal{C}_c$ consisting of 3 residual blocks that maintain the same spatial resolution and produce refined features $\{\tilde{\psi}^c\}_{C \times C \times C}$. These features are upsampled to a fine grid resolution using nearest-neighbor upsampling, and concatenated with fused features at fine level $\{\psi^f\}_{F \times F \times F}$. A fine-level 3D CNN $\mathcal{C}_f$ is then applied to the concatenated features, resulting in refined fine features $\{\tilde{\psi}^f\}_{F \times F \times F}$. Both, coarse $\tilde{\psi}^c$ and fine features $\tilde{\psi}^f$ are used for surface occupancy prediction.

**Coarse-to-fine surface filtering.** The refined features are also used to predict occupancy masks for near-surface locations at both coarse and fine levels, thus, filtering out free-space regions and sparsifying the volume, such that the higher-resolution and computationally expensive fine-scale surface extraction is performed only in regions close to the surface. To achieve this, additional 3D CNN layers $\mathcal{M}_c$ and $\mathcal{M}_f$ are applied to the refined features, outputting a near-surface mask $m^c, m^f \in [0, 1]$ for every grid point:

$$\mathcal{M}_c : \{\tilde{\psi}^c\}_{C \times C \times C} \mapsto \{m^c\}_{C \times C \times C}$$
$$\mathcal{M}_f : \{\tilde{\psi}^f\}_{F \times F \times F} \mapsto \{m^f\}_{F \times F \times F}$$

Only spatial regions where both $m^c$ and $m^f$ are larger than 0.5, i.e., close to the surface, are processed further to compute the final surface reconstruction; other regions are determined to be free space. This improves the overall reconstruction performance by focusing the capacity of the surface prediction network to close-to-the-surface regions and enables a significant runtime speed-up.

Intermediate supervision of near-surface masks $m^c$ and $m^f$ is employed using masks $m_{\mathrm{gt}}^c$ and $m_{\mathrm{gt}}^f$ generated from the ground truth scene reconstruction, denoting the grid point as near-surface if there exists ground truth surface in the radius of $v_c$ or $v_f$ from the point. Binary cross entropy losses $\mathcal{L}_c = \mathrm{BCE}(m^c, m_{\mathrm{gt}}^c)$ and $\mathcal{L}_f = \mathrm{BCE}(m^f, m_{\mathrm{gt}}^f)$ are applied.

## 3.3 Surface Occupancy Prediction

The final surface reconstruction is predicted by decoding the coarse and fine feature grids to occupancy values $o \in [0, 1]$, with values $o \geq 0.5$ representing occupied points and values $o < 0.5$ representing free-space points. For a point $\mathbf{p} \in \mathbb{R}^3$, we compute its feature representation by trilinearly interpolating coarse and fine grid features:

$$\psi_{\mathbf{p}}^c = \mathrm{Trilinear}(\mathbf{p}, \{\tilde{\psi}^c\}_{C \times C \times C})$$
$$\psi_{\mathbf{p}}^f = \mathrm{Trilinear}(\mathbf{p}, \{\tilde{\psi}^f\}_{F \times F \times F})$$

We concatenate the interpolated features and predict the point's occupancy as $o = \mathcal{S}(\psi_{\mathbf{p}}^c, \psi_{\mathbf{p}}^f)$, where $\mathcal{S}$ is a multi-layer perceptron (MLP) with 3 modules of feed-forward layers, containing ReLU activation, linear layer with residual connection, and layer norm.

**Surface occupancy supervision.** We train on $1.5 \times 1.5 \times 1.5$ m volumetric chunks of scenes for training efficiency. To supervise the surface occupancy loss, 1k points are sampled inside the chunk, with 80% of samples drawn from a truncation region at most 10 cm from the surface, and 20% sampled uniformly inside the chunk. Ground truth occupancy values $o_{\mathrm{gt}}$ are computed using the ScanNet RGB-D reconstructions [8]. For uniform samples it is straightforward to generate unoccupied point samples by sampling points in free space in front of the visible surface, but it is unknown whether a point sample is occupied when it lies behind seen surfaces. In order to prevent artifacts behind walls, we follow the data processing applied in [28] and additionally label point samples as occupied, if they are sampled in areas where an entire vertical column of voxels is occluded in the scene. A binary cross entropy loss $\mathcal{L}_o = \mathrm{BCE}(o, o_{\mathrm{gt}})$ is then applied to the occupancy predictions $o$.

### 3.4 View Selection for Online Scene Reconstruction

We aim to consider all $N$ frames as input to our transformer for each 3D location in a scene; however, this becomes extremely computationally expensive with long videos or large-scale scenes, which prohibits online scene reconstruction. Instead, we proceed with the reconstruction incrementally, processing every video frame one-by-one, while keeping only a small number $K = 16$ of measurements for every 3D point. We visualize this online approach in Fig. 1.

During training, for efficiency, we use only $K_t$ random images for each training volume. At test time, we leverage the attention weights $w^c$ and $w^f$ of the initial transformer layers to determine which views to keep in the set of $K$ measurements. Specifically, for a new RGB frame, we extract its 2D features, and run feature fusion for every coarse and fine grid point inside the camera frustum. This returns the fused feature and also the attention weights over all currently accumulated input measurements. Whenever the maximum number of $K$ measurements is reached, a selection is made by dropping out a measurement with lowest attention weight before adding new measurements in the latest frame. This guarantees a low number of input measurements, speeding up fusion processing times considerably. Furthermore, by using coarse-to-fine filtering, described in Sec. 3.2, we can further accelerate fusion by only considering higher resolution points in the area near the estimated surface. Together with incremental processing that results in high performance benefits, our approach performs per-frame feature fusion at about 7 FPS despite an unoptimized implementation.

### 3.5 Training Scheme

Our approach has been implemented using the PyTorch library [31]. The architecture details of the used networks are specified in the supplemental document. To train our approach we use ScanNet dataset [8], an RGB-D dataset of indoor apartments. We follow the established train-val-test split. For training, we randomly sample $1.5 \times 1.5 \times 1.5$ m volume chunks of the train scenes, sampling less chunks in free space and more samples in areas with non-structural objects, i.e. not only consisting of floor or walls. This results in $\approx 165$k training chunks. For each chunk, we randomly sample $K_t = 8$ RGB images among all frames that include the chunk in their camera frustums.

The 2D convolutional encoder $\Theta$ for image feature extraction is implemented as a ResNet-18 [14] network, pre-trained on ImageNet [24]. During training, a batch size of 4 chunks is used with an Adam [23] optimizer with $\beta_1 = 0.9$, $\beta_2 = 0.999$, $\epsilon = 10^{-8}$ and weight regularization of $10^{-4}$. We use a learning rate of $10^{-4}$ with 5k warm-up steps at initialization, and square root learning rate decay afterwards. When computing the losses of coarse and fine surface filtering predictions, a higher weight of 2.0 is applied to near-surface voxels, to increase recall and improve overall robustness. Training takes about 30 hours using an Intel Xeon 6242R Processor and an Nvidia RTX 3090 GPU.

## 4 Experiments

**Metrics.** To evaluate our monocular scene reconstruction, we use several measures of reconstruction performance. We evaluate geometric accuracy and completion, with accuracy measuring the average point-to-point error from predicted to ground truth vertices, completion measuring the error in the opposite direction, and chamfer as the average of accuracy and completion (in cm). To account for possibly different mesh resolutions among methods, we uniformly sample 200k points over mesh faces of every reconstructed mesh. Additionally, we threshold these point-to-point errors and compute precision and recall by computing the ratio of point-to-point matches within distance $\leq 5$ cm. Since it is easy to maximize either precision (by predicting only a few but accurate points) or recall (by over-completing reconstructions with noisy surface), we found the most reliable metric to be F-score, determined by both precision and recall.

Our ground truth reconstructions are obtained by automated 3D reconstruction [9] from RGB-D videos of real-world environments and, thus, they are often incomplete due to unobserved and occluded regions in the scene. To avoid penalizing methods for reconstructing a more complete scene w.r.t. the available ground truth, we apply an additional occlusion mask at evaluation.

As most state of the art, particularly for depth estimation, rely on a pre-sampled set of keyframes (based on sufficient translation or rotation difference between camera poses), we evaluate all approaches based on sequences of sampled keyframes, using the keyframe selection of [13].

Table 1: Quantitative comparison with baselines and ablations on test set of Scannet dataset [8].

| Method | Acc ↓ | Compl ↓ | Chamfer ↓ | Prec ↑ | Recall ↑ | F-score ↑ |
|---|---|---|---|---|---|---|
| RevisitingSI [18] | 14.29 | 16.19 | 15.24 | 0.346 | 0.293 | 0.314 |
| MVDepthNet [42] | 12.94 | 8.34 | 10.64 | 0.443 | 0.487 | 0.460 |
| GPMVS [17] | 12.90 | 8.02 | 10.46 | 0.453 | 0.510 | 0.477 |
| ESTDepth [26] | 12.71 | 7.54 | 10.12 | 0.456 | 0.542 | 0.491 |
| DPSNet [19] | 11.94 | 7.58 | 9.77 | 0.474 | 0.519 | 0.492 |
| DELTAS [38] | 11.95 | 7.46 | 9.71 | 0.478 | 0.533 | 0.501 |
| DeepVideoMVS [13] | 10.68 | **6.90** | 8.79 | 0.541 | 0.592 | 0.563 |
| COLMAP [37] | 10.22 | 11.88 | 11.05 | 0.509 | 0.474 | 0.489 |
| NeuralRecon [39] | **5.09** | 9.13 | 7.11 | 0.630 | **0.612** | 0.619 |
| Atlas [28] | 7.16 | 7.61 | 7.38 | 0.675 | 0.605 | 0.636 |
| Ours: w/o TRSF, avg | 7.23 | 9.74 | 8.48 | 0.635 | 0.501 | 0.557 |
| Ours: w/o TRSF, weight | 6.11 | 11.12 | 8.61 | 0.686 | 0.512 | 0.583 |
| Ours: w/o TRSF, conv | 6.56 | 9.84 | 8.20 | 0.661 | 0.524 | 0.582 |
| Ours: w/o spatial ref. | 10.46 | 16.91 | 13.68 | 0.479 | 0.295 | 0.361 |
| Ours: w/o C2F filter | 6.57 | 7.69 | 7.13 | 0.678 | 0.592 | 0.631 |
| Ours: w/o proj. depth | 8.06 | 10.02 | 9.04 | 0.594 | 0.475 | 0.525 |
| Ours: w/o viewing ray | 5.71 | 8.59 | 7.15 | 0.706 | 0.559 | 0.621 |
| Ours: 30 cm voxel size | 7.92 | 17.33 | 12.63 | 0.491 | 0.258 | 0.335 |
| Ours: 15 cm voxel size | 5.79 | 9.62 | 7.71 | 0.686 | 0.520 | 0.589 |
| Ours: 4 images, RND | 8.01 | 10.28 | 9.15 | 0.587 | 0.445 | 0.502 |
| Ours: 4 images | 6.80 | 8.40 | 7.60 | 0.661 | 0.524 | 0.581 |
| Ours: 8 images, RND | 6.74 | 8.55 | 7.64 | 0.665 | 0.544 | 0.596 |
| Ours: 8 images | 6.17 | 7.69 | 6.93 | 0.704 | 0.584 | 0.636 |
| Ours: 16 images, RND | 5.80 | 8.56 | 7.18 | 0.711 | 0.584 | 0.638 |
| Ours | 5.52 | 8.27 | **6.89** | **0.728** | 0.600 | **0.655** |

## 4.1 Comparison with State of the Art

In Tab. 1, we compare our approach with state-of-the-art methods. All methods are trained on the ScanNet dataset [8], using the official train/val/test split. We use the pre-trained models provided by the authors for MVDepthNet [42], GPMVS [17] and DPSNet [19] which are fine-tuned on ScanNet. For baselines that predict depth in a reference camera frame instead of directly reconstructing 3D surface, a volumetric fusion method [6] is used to fuse different depth maps into a 3D truncated signed distance field. The single-view depth prediction method RevisitingSI [18] suffers from the more challenging task formulation without the use of multiple views, leading to noisier depth predictions and inconsistencies between frames. Multi-view depth estimation methods leverage the additional view information for improved performance, with the LSTM-based approach of DeepVideoMVS [13] achieving the best performance among these approaches. Reconstruction quality further improves with methods that directly predict the 3D surface geometry, such as NeuralRecon [39] and Atlas [28]. Our transformer-based feature fusion approach enables more robust reconstruction and outperforms all existing methods in both chamfer distance and F-score. The performance improvement can also be clearly seen in the qualitative comparisons in Fig. 3.

## 4.2 Ablations

To demonstrate the effectiveness of our design choices, we conducted a quantitative ablation study which is shown in Tab. 1 and discussed in the following.

**What is the impact of learning to fuse features from different views with transformers?** We evaluate the effect of our learned feature fusion by replacing the transformer blocks with a multi-layer perceptron (MLP) that processes input image observations independently. The per-view outputs of this MLP are fused using an average (*w/o TRSF, avg*) or using a weighted average with weights

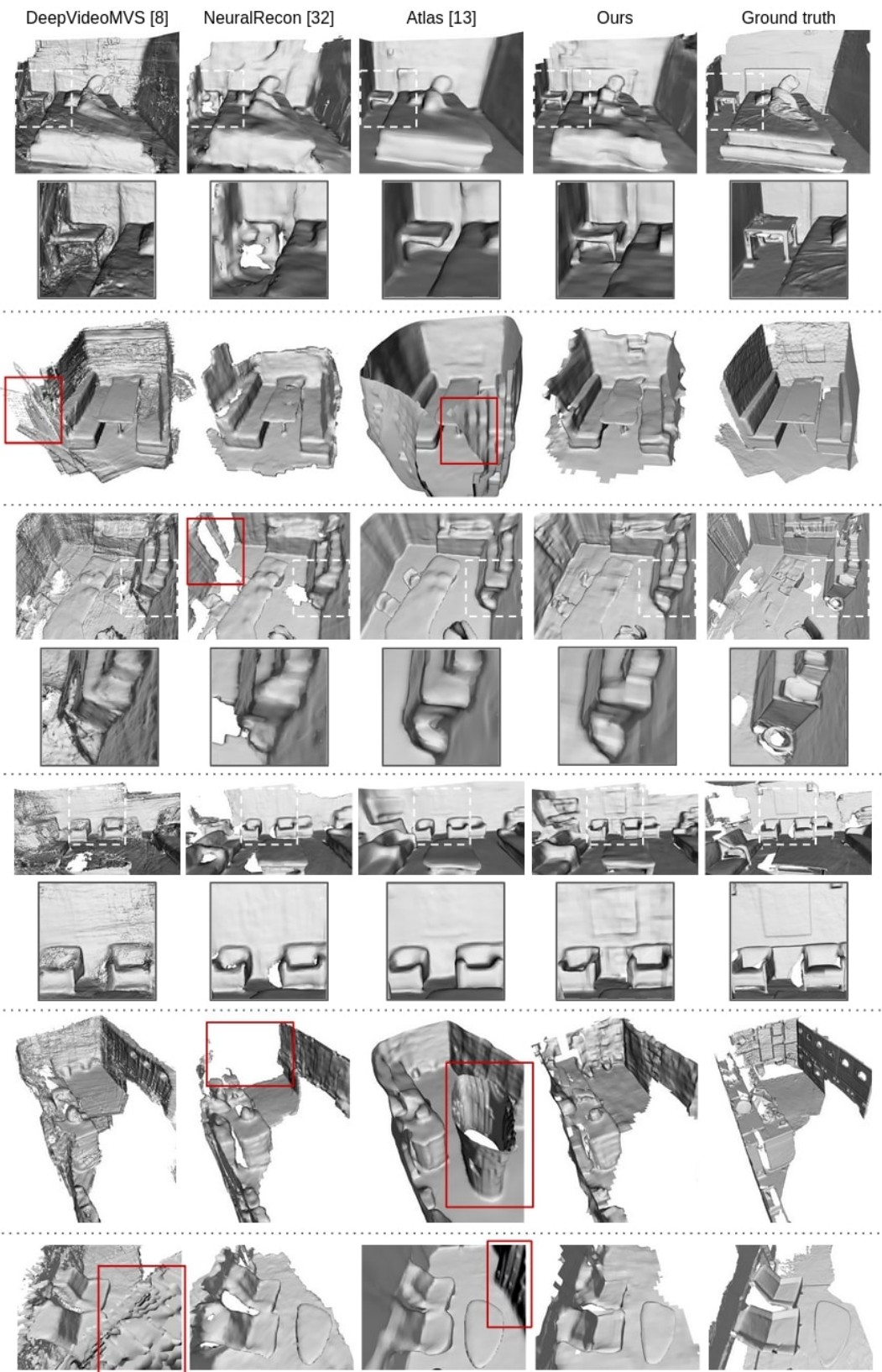

Figure 3: Qualitative comparison of scene reconstructions on test set of ScanNet dataset [8]; note that only RGB input is used by each method while the ground truth is reconstructed using the input depth.

predicted by the MLP (*w/o TRSF, weight*). Additionally, we implemented convolutional feature fusion, using a 1-dimensional CNN that processes features in temporal domain and predicts fused features (*w/o TRSF, conv*). We find that our transformer-based view fusion effectively learns to attend to the most informative views for a specific location, resulting in significantly improved performance over these feature fusion alternatives.

**Does spatial feature refinement help reconstruction performance?** Spatial feature refinement is indeed very important for reconstruction quality. It enables the model to aggregate feature information in spatial domain and produce more spatially consistent and complete reconstructions, without it (*w/o spatial ref.*) the geometry completion (and recall metric) are considerably worse.

**How important is coarse-to-fine filtering?** Predicting the coarse and fine near-surface masks provides an additional performance improvement compared to the model without it (*w/o C2F filter*), as it allows more focus on surface geometry. Furthermore, this enables a speed-up of the fusion runtime by a factor of approximately 3.5, resulting in processing times of 7 FPS (instead of 2 FPS).

**Are additional inputs to the transformer networks needed?** Existing reconstruction approaches [39, 28] aggregate 2D features using a simple average operation. In comparison, our approach uses a transformer to learn the feature fusion. That makes it possible to use additional inputs that don't support a straight-forward average operation, but could be very informative for the task of multi-view surface reconstruction, such as projected depth and viewing ray. In Tab. 1 we conducted an additional quantitative ablation study w.r.t. the input to the transformer networks. Both the projected depth as well as the view ray help the transformer to better fuse the features for the task of 3D reconstruction.

**How does voxel size of feature grids influence reconstruction performance?** We compared the reconstruction performance when using different voxel sizes for the feature grid. We only varied fine feature grid resolution, voxel size of coarse grid was always 30 cm. More specifically, we replaced the voxel size of 10 cm at the fine grid level with 30 cm and 15 cm. In both cases, the performance decreased considerably; i.e., the higher the resolution, the better the results. That is reflected also in qualitative comparison in the supplemental document.

**How many views should be used for feature fusion?** In our experiments, we use a limited number of $K = 16$ frame observations to inform the feature for every 3D grid location. We find that these views all contribute, with performance degrading somewhat with sparser sets of observations ($K = 8$ or $K = 4$). The number of frames is limited because of execution time and memory consumption for bigger scenes.

**How effective is frame selection using attention weights?** The $K$ frames for each 3D grid feature are selected based on the computed attention weights and are updated during scanning. To evaluate this frame selection, we compare against a frame selection scheme that randomly selects frames that observe the 3D location (*RND*), which results in a noticeable drop in performance for both chamfer and F-score. The performance difference is even larger when using less views for fusion ($K = 8$ or $K = 4$), where view selection becomes even more important. In Fig. 1, we visualize the most important view for locations in the scene, selected by the highest attention weight. Relatively smooth transitions between selected views among neighboring 3D locations suggest that view selection is spatially consistent. To illustrate the frame selection, we also visualize all selected frames with corresponding attention weights for specific 3D locations in the supplemental document.

## 4.3 Limitations

Under severe occlusions and partial observation of the scene, our method can struggle to reconstruct details of certain objects, such as chair legs, monitor stands, or books on the shelves. Furthermore, transparent objects, such as glass windows without frames, are often inaccurately reconstructed as empty space. We show qualitative examples of these failure cases in Fig. 4. These challenging scenarios are often not properly reconstructed even when using ground truth RGB-D data, and we believe that using self-supervised losses [10] for monocular scene reconstruction could be an interesting future research direction. Additionally, higher resolution geometric fidelity could potentially be achieved by sparse operations in 3D or learning local geometric priors on detailed synthetic data [21].

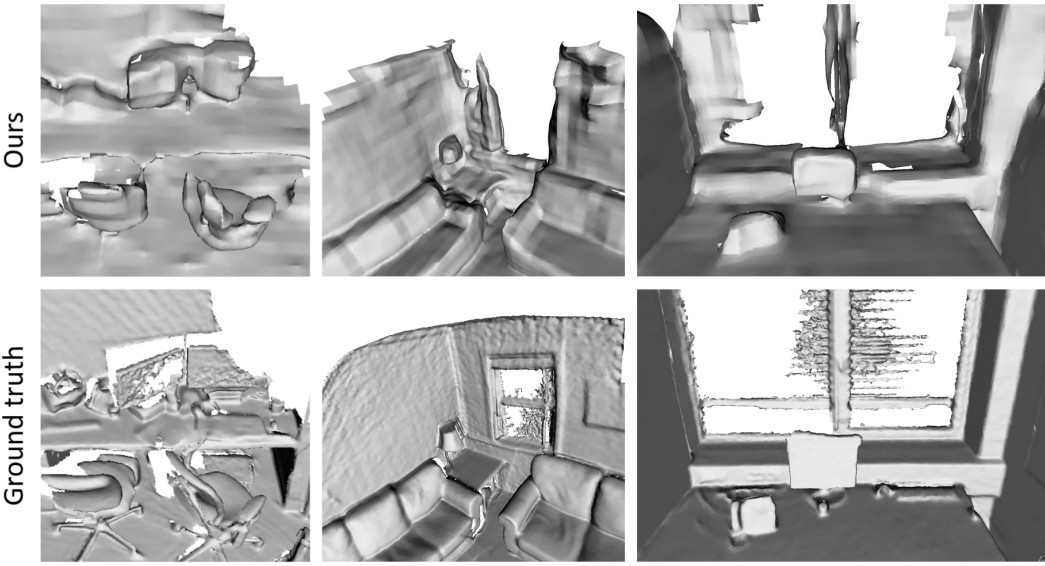

Figure 4: Limitations of our approach are the lack of detail at partially observed and occluded objects, and inaccurate reconstruction of transparent surfaces, such as glass windows.

## 5 Conclusion

We introduced TransformerFusion for monocular 3D scene reconstruction, leveraging a new transformer-based approach for online feature fusion from RGB input views. A coarse-to-fine formulation of our transformer-based feature fusion improves the effective reconstruction performance as well as the runtime. Our feature fusion learns to exploit the most informative image view features for geometric reconstruction, achieving state-of-the-art reconstruction performance. We believe that our interactive scanning approach provides exciting avenues for future research, and enables new possibilities in learning multi-view perception and 3D scene understanding.

## Broader Impact

Our work proposes a novel monocular scene reconstruction approach that can be used for applications in the field of augmented and virtual reality, and also serves as a basis for 3D scene understanding from monocular RGB input, enabling navigation of autonomous agents in unknown environments. Being a building block for these applications, we need to be aware of the potential negative societal impacts of some applications, such as the improper use of autonomous robots in military, or labor market disruptions as a consequence of job automation. Since our approach is data-driven, using RGB-D data as supervision, we also need to be aware of related privacy concerns when capturing new datasets for 3D reconstruction.

## Acknowledgments

This project is funded by the Bavarian State Ministry of Science and the Arts and coordinated by the Bavarian Research Institute for Digital Transformation (bidt), a TUM-IAS Rudolf Mößbauer Fellowship, the ERC Starting Grant Scan2CAD (804724), and the German Research Foundation (DFG) Grant Making Machine Learning on Static and Dynamic 3D Data Practical.

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
