# OpenReview forum: "TransformerFusion: Monocular RGB Scene Reconstruction using Transformers"
_NeurIPS.cc/2021/Conference — NeurIPS 2021 Poster_

### Official Review · Reviewer_AGwq · 2021-07-14

**Rating:** 6
**Confidence:** 4

**Summary:**

This paper presents an RGB 3D reconstruction algorithm with monocular RGB input, using transformer network. The transformer architecture is utilized for temporal feature fusion from multi-view images. Coarse-to-fine hierarchical scheme was designed for efficient surface detection and real-time reconstruction. Also, attention weights from the transformer is used in order to select the most dominant (feature-abundant) frames so that more interactive scene reconstruction can be available. By doing so, the proposed algorithm shows comparable performance on the benchmark experiment, compared to CNN-based previous methods.

**Ethical Concerns:**

.

**Limitations And Societal Impact:**

I appreciate the authors for mentioning the limitations of their algorithm on the lack of details in scene reconstruction results.

Additionally, considering the capture rate of the conventional RGB cameras, I do not think 7 FPS satisfies as ‘online’ or ‘real-time’.


**Main Review:**

This paper introduced a way of successfully utilizing transformer architecture inspired from NLP to the 3D scene reconstruction problem in computer vision. It is well-written and easy to follow. Also, the paper ablates well on their design choices.

The followings are my concerns and questions.
1. While I do think that utilizing transformer architecture in the 3D reconstruction field alone can be a meaningful contribution, the overall pipeline looks very similar to NeuralRecon [39], except for the substitution of GRU Fusion unit with the ViT-based transformer module. Please elaborate on the additional difference or stand-outs that the proposed algorithm has over [39]. It will be really helpful to reconsider my rating towards acceptance.

2. In figure 2, it seems like the coarse features are extracted from the 4th layer of the image encoder, and the fine features are from the 1st. Assuming the basic architecture of the ResNet, the former will have lower spatial resolution but the features will be more deeply embedded, and vice-versa. I wonder if the channel size of the encoded feature can also be a factor in choosing whether it is more appropriate for coarse/fine pipeline. What would happen if the 1st layer feature is utilized in T_c(Coarse Transformer) and the 4th layer feature is used for T_f(Fine Transformer)?


**Time Spent Reviewing:**

10

---

> ### Author Response · Authors · 2021-08-10
> **Reply to Reviewer AGwq**
>
> Thank you for your review and your questions!
>
> ### Relation to NeuralRecon
>
> Both our approach and NeuralRecon focus on online scene reconstruction from monocular video. However, while NeuralRecon demonstrates impressive online performance, it performs a two-step reconstruction of local chunks followed by a global scene fusion, and aggregates 2D features by averaging. In contrast, our transformer-based fusion enables attending to the most informative features from the 2D images, resulting in improved reconstruction performance (check Table 1 in the main paper). This additionally enables adaptive frame selection for robust online reconstruction.
>
> *Temporal attention for feature fusion.* The local chunk reconstruction of NeuralRecon follows existing approaches (e.g., Atlas), which aggregate 2D features along different views using a simple feature average operation to compute a fused chunk feature for every 3D voxel. The novelty of our approach is that the fusion of 2D features from different views is learned using a transformer network in self-supervised fashion. This offers additional flexibility to the network, it is able to **recognize more valuable observations and filter out pixels under motion blur, occlusions**, etc. Furthermore, we **can append additional inputs** to the network, such as camera view ray, distance to camera center. In comparison to NeuralRecon these properties do not have a straight-forward average operation, but are very informative for the task of multi-view surface reconstruction. It is also important that these observations are processed jointly via a transformer, since it’s **hard to independently argue what a good observation for multi-view reconstruction is** (e.g., for two observations can be good if there is a large enough translation motion between them).
>
> *Online reconstruction.* To make the method run online, NeuralRecon considers only 9 temporally close frames for chunk reconstruction. It’s often challenging to reconstruct local chunks robustly by using only temporally close frames, since **further away observations can add more stable epipolar constraints**. To achieve global scene reconstruction, they require a different module that fuses different chunk reconstructions, here GRU-based fusion is employed. However, at this stage it’s hard to correct reconstruction errors resulting from too-local frame selection, the task of GRU fusion is mostly combining existing chunk reconstructions. In our approach, we do not need a separate module/step for global fusion, since the **transformer can work efficiently with global observations**: at train time, these are generated randomly, and at test time the most important observations are selected via transformer attention outputs. That simplifies the reconstruction approach, relying on the transformer network that can efficiently aggregate global information from the entire sequence, no matter how long the sequence is. Our method therefore doesn’t suffer from error accumulation from local chunks, since **views are fused globally and selected depending on their importance**, not just temporal closeness.
>
> ### Ablation Study w.r.t. the used Image Features
>
> As an ablation study on the used image features, we tried the suggested configuration: using coarse image features for the fine voxels and the fine image features for the coarse voxels. This leads to significantly worse results, reducing the F-score to 0.536 (c.f., 0.655 for the proposed feature usage).
>
> ### Limitation
>
> The capture rate of a conventional RGB camera is higher than our reported 7fps. Our approach does not rely on a dense observation of frames, but can operate on a subset of frames (also called keyframes). All experiments in our paper are run on keyframes. Thus, a reconstruction (as shown in the video) can be done in an online fashion. Under very fast camera motions, such keyframing could potentially miss some information;  we will clarify this limitation in the paper.

---

> > ### Comment · Reviewer_AGwq · 2021-08-27
> > **Response to the authors' rebuttal**
> >
> > I thank the authors for their in-depth comparison between this paper and NeuralRecon, providing much more insights on the advantage of utilizing a transformer network.
> >
> > I see that all other reviewers are also pointing out the lack in technical novelty, especially comparing it to NeuralRecon.
> > Although the overall pipeline is still similar to previous works, I believe that this paper can be credited for its analysis on what can we expect from using a transformer network in 3D reconstruction field.
> >
> > Therefore, I change my initial rating to more positive decision.

---

### Official Review · Reviewer_4TZ2 · 2021-07-16

**Rating:** 6
**Confidence:** 4

**Summary:**

This paper proposes a method to reconstruct dense surface geometry of a static scene from a monocular RGB video.

The method combines implicit functions and transformers for achieving the final reconstruction. Specifically, images in the video are encoded by a 2D image encoder. After that, for each voxel in a 3D feature volume, the feature was calculated by: first bilinearly sampling per-view features by projecting the voxel on the 2D feature maps in different views, and then, merge the per-view features from different views together using a multi-head transformer. After calculating the 3D feature volume, 3D convolutions are used to directly refine the features in it. Finally, for each query point in 3D space, an MLP-based occupancy decoder was introduced to decode the occupancy value given the corresponding feature of the query point (which is sampled in the refined 3D feature volume).
To further improve the reconstruction efficiency, the whole reconstruction pipeline was conducted in a coarse-to-fine manner.

The main contribution of the paper lies in the transformer-based feature merging stage. By utilizing the strong self-attention performance in the transformers for aggregating features from different views, the proposed method achieves plausible and more complete surface reconstruction results than previous methods.

**Ethical Concerns:**



**Limitations And Societal Impact:**



**Main Review:**

Reconstructing dense surface geometry using a monocular RGB camera is a very challenging task. Accurate and complete surface reconstruction will benefit a lot of areas like AR/VR, semantic understanding, etc.

The paper writing is clear and easy to follow. The results are plausible, and the experiments are thorough.

My major concern is the technical novelty of the proposed method. Since there already have a lot of works that use transformers for feature aggregation (as mentioned in the related work section), simply transferring this technique to static scene reconstruction is not that a breakthrough, despite the fact that the way of using a transformer is relatively straightforward in the paper.
Moreover, as shown in Fig.1 (1st row) of the supplementary material: the blue color views share similar image quality and viewing direction with the red color views. However, they contribute quite differently to the final results according to the output of the transformer, which is confusing. I would like to suggest the authors explain the reason for this phenomenon in detail.

**Time Spent Reviewing:**

3

---

> ### Author Response · Authors · 2021-08-10
> **Reply to Reviewer 4TZ2**
>
> Thanks for reviewing our paper! In the following, we discuss the concerns you raised in the review.
>
> ### Technical Novelty
> In our paper, we introduce and analyze the first transformer-based monocular 3D reconstruction method for large-scale scenes. The application of a transformer network to this scenario is non-trivial and needs a carefully crafted method. In our experiments, we show the influence of each of the required components to achieve the shown state-of-the-art reconstruction results.
>
> ### Contribution of views (Fig.1, suppl. document)
> There are different colors (and, thus, attention weights) for similar views. We observed that the attention head in the transformer architecture tends to sparsify the views, assigning a high weight to a single view, and low weights for the rest, as other similar views tend to provide only redundant information. Such behavior is well-suited for the task of reconstruction, where multiple different observations are more useful, especially ones observed under **different viewpoints and camera translations**. This is achieved by using multiple attention heads (in our architecture we use 8 heads), each specializing for a certain view type, and each picking only a single view representative. This leads to elimination of very similar views that are redundant, and at the same time encourages high weights for different views that are more useful -- additionally, this enables the attention weights to be very effective for online view selection. It is also a notable difference to NeuralRecon, where all views are treated the same (by averaging over view features).

---

### Official Review · Reviewer_Ert2 · 2021-07-17

**Rating:** 7
**Confidence:** 4

**Summary:**

The paper presents a fully end-to-end trainable deep 3D reconstruction pipeline taking monocular RGB video sequences as input. The proposed pipeline works in an online fashion and at interactive frame rates.
In the first stage, the method extracts 2D features from input RGB images. In a next stage, 3D features are generated by projecting a 3D location into the images, pooling the corresponding 2D features and fusing them through a transformer network. After refining the 3D features with 3D convolutions, the geometry is predicted through a neural implicit decoder and finally extracted through marching cubes.
Notably, the method uses a two-level hierarchy of features for improved quality and runtime performance. Furthermore, the self-attention weights are reused to only keep the most important observations for each 3D point at any given time.
Finally, the authors compare their pipeline to several state-of-the-art methods and perform an ablation study to validate the individual components.


**Ethical Concerns:**

No ethical concerns.

**Limitations And Societal Impact:**

The potential negative societal impact has been addressed adequately.
Limitations have been addressed somewhat. For me it is not clear yet where the lack of details in qualitative results stems from. Is it only due to occlusions? Could it be that the refinement step (3D convolution) smooths out a lot of details? Could this be addressed with another self-attention mechanism in the refinement step?


**Main Review:**

#Originality:
The idea of integrating self-attention into online 3D reconstruction pipelines and using the attention weights for view selection is novel and interesting, although not ground-breaking.
Using feature hierarchies or coarse-to-fine level approaches in order to reduce run-time or improve quality by incorporating larger perception fields is a well tested concept (e.g. Chibane et al. [5]; Saito et al. 2020 PIFuHD; NeuralRecon [39]) and not really new in the field of 3D reconstruction.
As such, this paper can be considered more a work which combines well-known techniques (Self-attention, Image Feature Extraction with 2D convolutions, using Feature Hierarchies, 3D Convolutions for “Refining”/Propagating 3D Features and using neural implicit functions to predict geometry) rather than a theoretical contribution but is nonetheless valuable and interesting.
In contrast to previous work the presented method is online and fast enough to run at interactive frame-rates. There exist other deep learning based online 3D reconstruction approaches (e.g. RoutedFusion, Weder et al.) but they work on depth inputs which is an easier task than RGB-only input.

#Quality:
From a qualitative perspective, the work feels well thought through and complete and the paper makes sense from a technical standpoint. The performed experiments and evaluations are extensive and one can see that the authors put a lot of effort into establishing a fair comparison with multiple SotA methods and baselines. The ablation study is very informative and valuable and proves that every proposed component is useful. Specifically, during the read-through, I was asking myself how much influence the viewing ray input had and was glad to find the corresponding ablation study in the supplementary material.

#Clarity:
The paper is well written, is clear and well organized. The most relevant details are present and the pipeline should be reproducible with help of the supplementary material. The most important related work has been adequately cited.

#Significance:
Quantitatively, the paper only slightly outperforms previous work (Atlas) and is roughly on par with SotA (NeuralRecon). Qualitatively, they seem to yield better results, even better than NeuralRecon. Therefore, the results are interesting and show that at least some level of improvements can be expected when using self-attention. I believe that others will build upon the idea to incorporate self-attention into deep learning architectures for 3D reconstruction pipelines and therefore think that exploring self-attention for 3D reconstruction pipelines is valuable. Besides the quantitative results being not overly impressive, I believe that the results provided are valuable to the community.

#Post-rebuttal:
I agree with the findings of reviewer mh3Q that the novelty of the paper is of an incremental nature. Nevertheless, I believe that the improved results and the technical contributions offset this and justify accepting the paper.
NeuralRecon can be viewed as parallel work, given that it has been published at CVPR 2021 (after NeurIPS deadline) and therefore I consider coincidental similarities in the pipelines not as a valid reason for rejecting the paper (although in my opinion, there are enough technical differences in the two pipelines, e.g., TransformerFusion projects 3D points into views, while NeuronRecon uses raycasting).

The observations of reviewer 4TZ2 regarding Fig. 1 of the supplementary were concerning at first but the answer of the authors makes sense, i.e. that out of several similar views only one is assigned the highest weight in order to remove redundancy. However, I suggest, the authors include their answer in the paper and/or the supplementary.

For the above mentioned reasons, my initial rating persists.

**Time Spent Reviewing:**

6

---

> ### Author Response · Authors · 2021-08-10
> **Reply to Reviewer Ert2**
>
> Thank you for the review and the helpful comments! Below, we answer your questions.
>
> ### Originality
> While we draw inspiration from other methods (e.g., self-attention used in NLP or image classification), we design and propose the first transformer-based 3D reconstruction method from monocular videos and analyze how a transformer-based fusion of features drawn from 2D can be leveraged for 3D reconstruction. This is particularly well-suited for 3D reconstruction from monocular videos, allowing our transformer to attend to more important features from different input observed frames.
>
> ### Significance
> We see a significant difference in the qualitative comparison. For quantitative evaluation, we outperform state of the art in F-score (0.655), while Atlas (the best offline method) achieves 0.636 and NeuralRecon (the best online baseline) results in 0.619. Here, we consider the F-score as the main evaluation metric, since Prec (precision) and Recall can be treated against each other (i.e., if you predict only geometry in a region where you are very certain, you get a high precision, but a low recall score and vice-versa if you predict geometry everywhere you get a high recall score, but a low precision), thus, a better score in only one of these measurements does not necessarily indicate better reconstructions.
>
> ### Limitations
> We agree that there is still room for improvement in terms of the reconstruction of details. Replacing the 3D convolutional network for the refinement with a self-attention mechanism is an interesting idea which could improve the quality further. We are happy to add further discussion.

---

### Official Review · Reviewer_mh3Q · 2021-07-18

**Rating:** 6
**Confidence:** 4

**Summary:**

This paper proposes to use vision Transformer in a coarse-to-fine manner as a new learned multi-view feature fusion scheme in monocular reconstruction task. Additional blocks such as spatial refinement and C2F filtering help the method generate coherent geometry and improve computational efficiency.

Compared to existing approaches, the full TransformerFusion system shows competitive results on 3D reconstruction quality using ScanNet benchmark with detailed ablation study.

**Limitations And Societal Impact:**

The authors have well addressed the potential negative societal impact of their work in the main paper.

**Main Review:**

There are many hanging fruits of applying Transformer in vision tasks. In this paper, the idea of using Transformer to adpatively select which frames to contribute more is well motivated and technically sound. As vision transformer naturally fits the task of temporal fusion of sequential video image data. Both quantitative and qualitative results shows the effectiveness of proposed system.

Pros:
 - Paper is well written and the ablation experiments address several concerns of architecture design.
 - The introduction of transformer in multi-view feature fusion is well motivated.
 - The carefully designed pipeline reach competitive performance with reasonable computational cost and experiments are  well-designed.



Cons:
- Incremental contribution.

    - Given the general pipeline of TransformerFusion is similar to existing approaches such as NeuralRecon, the claimed two unique contributions of this paper come from the adoption of Transformer and C2F refinement to make Transformer practically applicable. In my understanding, these two contributions is in principle a single contribution, i.e., bringing Transformer into monocular reconstruction and make it applicable. The contribution is a bit weak and incremental from this point of view.

    - As mentioned above, the designed modules look very tailored to Transformer-based fusion. To strengthen the contribution and its generalisation, though not the main focus of this paper, it would be good to see if proposed modules like C2F surface filtering is beneficial to other exisiting systems as well.

- Data Generalisation

    - Current evaluation is tested using ScanNet alone and a common concern remains on the generalisations of trained TransformerFusion to other unseen scenes like 7-scenes or real-world exemplar scenes.

- Numerical Comparison

    - From Table 1 and Figure 2 in supplementary material, the refinement modules shows the strongest influence to reconstruction performance, even larger than the contribution from Transformer itself. Given the noisy qualitative result without refinement, is it caused by the lack of inductive biases in MLP-based Transformer fusion?

    - Based on previous point, in Table 1 author compares the ablation experiments without Transformer fusion but a plain MLP with learned or equal weights during fusion. For me it is more interesting and fair to see how convolutional-like fusion structure performs here. In addition. given inductive biases in convolutional structure, will the benefits from refinement module become less pronounced?

    - More explanation of these factors would be appreciated and help better understand what is the pros and cons of using Transformer here.

Some detailed comments and questions:

(1) How to select frames in video at test time? Are all frame-rate frames fed into the network or a pre-sampled subset is input due to efficiency consideration?

(2) An implicit MLP decoder is used to finally predict the occupancy values. However, as mentioned in the limitation,
fine structures are prone to missing which is a slightly against the advantages of using continuous MLP. Is this result related to the pre-defined voxel resolution in coarse and fine scale or due to the limited resolution of fused feature?

(3) In section 3.1, a 3D point in world coordinate is projected to multiple camera to query 2D features, is depth information of each view/point used here during projection? Counterpart approaches such as Altas or NeuralRecon emit a ray traversing the whole 3D voxel as no depth information is given. Could the author clarify this?

(4) How about using more than 16 frames as only fewer frames are tested. Is there a trend to saturate the performance given more views and redundancy or the performance consistently improves?

Overall, I think this is a well-motivated paper and is well written, therefore tend to accpet this paper (between 6 and 7). The quality of this paper is technically sound, though the contribution is a bit incremental and weak in my point of view, considering that applying Transformer to this task is the key uniqueness. Transformer shows its merits in multi-view fusion but there also comes with other related issues such as lack of inductive bias, computational cost due to point-wise query. More in-depth analysis of the pros and cons of using transformer in mono recontruction is appreciated, especially compared with other learning-based fusion approach using convolutional structure.



**Time Spent Reviewing:**

6 hours

---

> ### Author Response · Authors · 2021-08-10
> **Reply to Reviewer mh3Q**
>
> Thank you for the detailed comments and suggestions!
>
> ### Novelty / Contribution
> We introduce a transformer-based approach for large-scale 3D scene reconstruction from monocular video. In particular, we design our reconstruction to leverage a transformer for temporal feature fusion, which is particularly well-suited for the task of 3D reconstruction, where attending to more informative frames enables more robust reconstruction, with flexibility to reduce bias in comparison to feature averaging (which can tend to regions that have been seen by more frames), attend to image features from frames with less motion blur or occlusion, etc. The learned attention weights further enable an efficient online reconstruction through adaptive view selection. The coarse-to-fine surface filtering indeed shares some similarity with NeuralRecon, and it would be interesting to further explore how beneficial it could be to other existing systems. Our approach demonstrates this effectiveness in a combination with transformer-based temporal feature fusion with coarse-to-fine surface filtering, thus sparsifying regions of interest in spatial domain, while the latter efficiently selects most important observations in the temporal domain.
>
> ### Data Generalization
> We additionally evaluate our approach on TUM RGB-D sequences, featuring office-like environments. These sequences were captured with a Kinect sensor, while the ScanNet data that we used for training was recorded using a StructureIO sensor that uses an iPad RGB camera. Our approach achieves an F-score of 0.437, which is less than on the ScanNet test set but maintains accurate capture of scene structure. We believe the difference is mainly caused by a different RGB sensor used at test time.  We note that in a practical scenario, one would fine-tune on the respective test device’s camera and/or include recordings from multiple different sensors. We would be happy to include a detailed study in the final paper.
>
> ### Numerical Comparisons
> Due to efficiency reasons, we run feature fusion using transformers only in the temporal domain, so our spatial CNN refinement helps to handle aggregating information in the 3D spatial domain. This 3D spatial refinement is crucial for spatially-consistent surface prediction, and a 3D CNN is a natural architecture for spatial refinement, possibly also because of its inductive bias.
> However, if we use a CNN network for temporal fusion as well, replacing our transformer architecture with 1D temporal convolutions, this results in considerably worse reconstruction performance, achieving an F-score of 0.582. This is comparable to the fusion with an MLP with predicted weights (F-score of 0.583), while our transformer-based fusion achieves a higher F-score of 0.655. We believe the reason for this large improvement is the ability of our transformer design to more efficiently aggregate global information in temporal domain, which is often more important than local information (e.g., frames with larger motion between them can be more useful for geometry estimation, due to more numerically stable epipolar constraints, compared to locally close frames with small camera motion).
>
> ### Q&A
> 1. At test-time, we use a subset of frames, following the keyframe selection approach described in DeepVideoMVS [Düzçeker et al. 2020]. This is done due to efficiency considerations, to be able to execute reconstruction in an online fashion (the fusion runs at 7FPS, which matches the average keyframe sampling of 5-7 FPS). All experiments were done with this subset of frames.
>
> 2. In our experience, the continuous MLP representation is indeed good at representing surface details: when overfitting to a single Scannet sequence from the training set, most detail can be retrieved, so the representation power of the current architecture is quite high. But when trained on a complete training set for generalization, the network can sometimes struggle to recover some detail on validation sequences. As you suggested, a larger feature size, but potentially also a larger architecture, might result in a better generalization performance. However, more compute power would be required, and to cope with overfitting, we might need to enlarge our training corpus as well.
>
> 3. No depth is used during projection or in any other way during inference;  ground truth depth is used only during training and only for loss supervision. Similar to NeuralRecon/Atlas, a dense grid is defined such that it covers all camera view frustums. So all voxels in visible space are projected and their corresponding features are queried, with most of them being free-space voxels. The projection is done in the direction from voxel centers to image features (instead of the opposite direction, pixel rays to voxels) mainly for efficient computation and easier gradient back-propagation. It’s the task of the network to produce surface estimates, starting from a dense but coarse grid (with a lot of free-space voxels), and gradually sparsifying it with higher resolution evaluation around the predicted surface area.
>
> 4. Currently, we have used at most 16 views; however, we will be happy to include an ablation using more than 16 views.

---

> > ### Comment · Reviewer_mh3Q · 2021-08-25
> > **Reply to  Authors' Response**
> >
> > Thank the authors for the reponse. After reading the authors' responce and other reviewer's comments, most of my concerns are well addressed.
> >
> > Detailes feedbacks are shown below:
> >
> > - **About significance**. Most of the reviews show their concerns to the incremental novelty of this submission which uses Transformer for temporal multi-view feature fusion in monocular reconstruction task.  I acknowledge the nolvelty of this idea but feel it less supportive to stand alone.
> > As mentioned by second paragraph in my previous 'Incremental contribution' comment  and Reviewer Ert2, the Transformer as well as other modules such as surface filtering, refinement, intiail attention weight for view selection are bundled together which leads to the final performance. The improved description of contribution and more investigation of applying these modules to other pipeline can be strengthened in the paper and future work.
> >
> > - **Data Generalization and Numerical Comparisons**.
> > Thank the authors to conduct extra experiments on TUM datasets and the results show that it works reasonbly well under variontion of camera intrinsics and capturing devices. The abaliation study of temporal fusion using CNNs is adequate to show the advantage of Transformer on this task.
> >
> > - **MLP**.
> > Add more discussion and exploration on its incapability to catch fine structures is quite meaningful. In addition to the MLP structure, will the pre-defined voxel resolution at coarse and fine scales largely affect the results?
> >
> > Overall, I think most of the concens are well addressed by the author response. Although I have some slight concern on the significance, I think this submission is interesting and can be good to the area on multi-view visual information fusion and reconsturction.
> > Therefore, I personally tend to accept this submisison and keep my original positive score.

---

> > > ### Author Response · Authors · 2021-09-08
> > > **Additional reply to Reviewer mh3Q**
> > >
> > > Thank you for your reply, and useful feedback!
> > >
> > > We ran an ablation with respect to the resolution of the feature grids. More specifically, we replaced the original voxel size of 10cm at the fine grid level with 30cm and 15cm. In both cases, the performance decreased considerably; i.e., the higher the resolution, the better the results. We obtain an F-score of 0.335 for the voxel size of 30cm and F-score of 0.589 for the voxel size of 15cm. Our approach with a voxel size of 10cm achieves F-score of 0.655. In principle, we could even further increase the resolution; however, we are currently using only a single GPU for training. Overall, voxel resolution indeed influences the reconstruction of finer details; we will add some qualitative comparisons in the final paper.
> > >
> > > | Voxel size | F-score |
> > > |------------|---------|
> > > | 30 cm      |   0.335 |
> > > | 15 cm      |   0.589 |
> > > | 10 cm      |   0.655 |

---

### Author Response · Authors · 2021-08-10
**General Reply**

We would like to thank all reviewers for their detailed feedback!

We are glad that all reviewers agree that the method is technically sound, that the paper is ‘well written and organized’ [mh3Q,Ert2,4TZ2,AGwq], ‘easy to follow’ [4TZ2,AGwq] and the experiments are ‘thorough’, ‘well designed’[mh3Q], and ‘ablates well on the design choices’ [AGwq,Ert2]. The usage of a ‘transformer in multi-view feature fusion is well motivated’ [mh3Q], ‘novel and interesting’ [Ert2] and the pipeline is ‘carefully designed’ to ‘reach competitive performance’ [mh3Q].

We propose the first reconstruction method from monocular video that leverages transformers to obtain a detailed 3D reconstruction. In particular, we show that multi-head attention is ideally suited to fuse (and weight) features from different input frames in a coarse-to-fine fashion, thus enabling higher quality reconstruction than to state-of-the-art methods and enabling online reconstruction without requiring access to all frames in a sequence (or window) ahead of reconstruction time.

Specifically, the online reconstruction method NeuralRecon [Sun et al. 2021] averages features from different views without any difference in weight for different views. It is also a temporally local, chunk-wise reconstruction method, while our approach fuses features globally with attention-based weighting -- this leads our approach to achieve better reconstructions, both qualitatively and quantitatively. Our online method also shows considerable reconstruction quality improvement compared to the offline reconstruction method ATLAS [Murez et al. 2020].

We encourage the reviewers to also read our responses to the other reviewers.

Thank you,

the authors

---

### Decision · Program_Chairs · 2021-09-27

**Decision:**

Accept (Poster)

**Comment:**

This paper has rather positive reviews (7,6,6,5). In general, the reviewers appreciated the quality of the writing and experiments, as well as the (moderate) novelty the paper offers. Reviewer AGwq in particular was convinced by the rebuttal, especially on novelty vs the recent NeuralRecon paper, and increased their score from 5 to 6. Overall, the AC agrees with the reviewers and recommends acceptance.